# The Hybrid System for the Magnetic Characterization of Superparamagnetic Nanoparticles

**DOI:** 10.3390/s22228879

**Published:** 2022-11-17

**Authors:** Mateusz Midura, Przemysław Wróblewski, Damian Wanta, Jacek Kryszyn, Waldemar T. Smolik, Grzegorz Domański, Michał Wieteska, Wojciech Obrębski, Ewa Piątkowska-Janko, Piotr Bogorodzki

**Affiliations:** Institute of Radioelectronics and Multimedia Technology, Warsaw University of Technology, 00-655 Warsaw, Poland

**Keywords:** complex magnetic susceptibility, superparamagnetic nanoparticles, AC magnetometry, hyperthermia therapy

## Abstract

The characterization of nanoparticles is crucial in several medical applications, such as hyperthermic therapy, which heats superparamagnetic nanoparticles with an external electromagnetic field. The knowledge of heating ability (magnetic losses) in AC magnetic field frequency function allows for selecting the optimal excitation. A hybrid system for the characterization of superparamagnetic nanoparticles was designed and tested. The proposed setup consists of an excitation coil and two sensing probes: calorimetric and magnetic. The measurements of the imaginary part of the complex magnetic susceptibility of superparamagnetic nanoparticles are possible in the kilohertz range. The system was verified using a set of nanoparticles with different diameters. The measurement procedure was described and verified. The results confirmed that an elaborated sensor system and measuring procedures could properly characterize the magnetic characteristics of nanoparticles. The main advantage of this system is the ability to compare both characteristics and confirm the selection of optimal excitation parameters.

## 1. Introduction

Superparamagnetic nanoparticles have important medical applications, both in diagnostic imaging and therapy. They can be used in magnetic resonance imaging (MRI) as contrast enhancement agents, or in magnetic particle imaging (MPI) as tracers for vascular and cancer diagnostics [1]. Promising therapeutic methods are targeted drug delivery [2] and magnetic nanoparticle hyperthermia [3]. Understanding the magnetic nanoparticles’ physical and chemical properties is critical in these applications. Characterization of nanoparticles includes measuring their diameter (core and hydrodynamic) and determining the chemical and magnetic properties of the coating and core [4].

In hyperthermia treatment using magnetic nanoparticles (MNPs), magnetic characterization can be used for maximization of the losses in nanoparticles in the function of magnetic field excitation frequency [5], whereas the energy dissipated in the rest of the patient’s body due to the Eddy currents might be kept at a reasonably low level [6].

Magnetic characterization can be conducted using either magnetic [7,8] or calorimetric measurements [9,10]. In recent studies, different types of AC magnetometers were presented and tested using superparamagnetic nanoparticles [11,12]. Geraio et al. [13] proposed a system consisting of a single-layer excitation coil with a hollow copper pipe used as a wire. The setup was able to measure magnetic losses by integrating a magnetic hysteresis loop in a range of 49 kHz to 1030 kHz. Another benchtop magnetometer was presented by Saari et al. [14]. The authors wound a small excitation coil with Litz wire and tested different designs of two oppositely wound detection coils that allowed for measurements in the range from 5 Hz to 158 kHz. Magnetic susceptibility characteristics for very low frequencies were also achieved using a Helmholtz excitation coil by Kuipers et al. [15] and Saari [16]. Wu et al. in [17] presented a different approach for nanoparticle characterization using the frequency-mixing method. Two excitation coils were combined in one measurement system to generate a magnetic field at two different frequencies simultaneously. In other studies, magnetic characterization of nanoparticles was performed using the calorimetric method. The use of a commercial heating device with an adjustable magnetic field frequency range from 150 kHz to 400 kHz was reported in [18,19]. Some researchers conducted the experiments using self-designed adiabatic magnetothermal setups [20,21].

Despite the potential importance of magnetic nanoparticles in biomedicine, no standardized procedures for measuring their physical properties are available. There is a need to develop standard operating methods for magnetic measurements, calibrated measurement devices, and certified reference materials [22,23,24]. 

The aim of this work was to design a hybrid system that allows for measuring the magnetic susceptibility of superparamagnetic nanoparticles using two different methods: calorimetry and AC magnetometry. To the best of the authors’ knowledge, no other setup combines both methods in one system. The excitation module in the presented hybrid system is the same for both measurements. Two detection probes can be used interchangeably. According to the studies [25,26,27], we chose a range of excitation frequencies from single kHz to a few hundred kHz. This frequency range is sufficient to observe magnetic heating losses due to Brown relaxation generated in the nanoparticles with a diameter from 15 to 30 nm. The local optimal heating frequency in a low-frequency range (OHF_LF_) (from 4 kHz to 382 kHz) for hyperthermic treatment can be determined based on characteristics of the imaginary part of magnetic susceptibility. We have presented an innovative procedure based on both methods, which allows for faster OHF_LF_ determination than would be possible on the basis of calorimetric measurements only.

## 2. Materials and Methods 

The elaborated system consists of an excitation coil and two sensing probes, calorimetric and magnetic, that can be used interchangeably (Figure 1). It was assumed that measurements of the imaginary part of complex magnetic susceptibility could be conducted in the range of 4 kHz to 382 kHz and at a magnetic field up to 15 mT. Such a value of magnetic field requires a large current flow through the excitation coil. That could be achieved by applying a series resonant circuit, in which the impedance is small enough to draw large amounts of current from the amplifier. A set of different value capacitors is required to ensure a measurement in the above-specified frequency range. In a series resonant circuit, sixteen different values of capacitances were applied interchangeably. Due to the high voltage (max. 10 kV at 382 kHz) generated in this circuit, a parallel-series connection of capacitors was used for each capacitance value.

### 2.1. Excitation Coil Design

The excitation coil has been designed as a double-layer coil with 122 turns, wound at a distance of 200 mm (Figure 1a). In some studies [13,15,28], a single-layer excitation coil was used. Authors argue that one layer provides the best energy efficiency in magnetic field generation [28]. However, this analysis does not consider the maximum ampacity of the wire, especially if Litz wire with a large number of thin single strands has been used. If a limitation for maximal current through the wire is also considered, then in some cases winding two layers could result in a higher magnetic field. At higher frequencies, high voltage up to a few kilovolts between the layers becomes a significant problem, as well as the self-resonant frequency that is much lower in comparison with single-layer coil, due to an additional interlayer component of parasitic capacitance created between neighboring layers. However, interlayer parasitic capacity can be reduced by changing the distance between the adjoining layers. In our design, 3.5 mm round wooden stick spacers have been used to separate the layers. Figure 2 shows inductance, and impedance spectroscopy for excitation coil composed of two tightly adjacent layers, and the same coil with two layers separated from each other at a fixed distance of 3.5 mm. There is a significant increase in self-resonant frequency due to a remarkable reduction in an interlayer’s parasitic capacitance. Little increase of coil resistance, due to a bigger radius of turns in the second layer, was also noted. An air gap between layers is also necessary to ensure sufficient breakdown voltage between the first and last turn that could generate voltage up to few kV. 

Sixteen different values of capacitors were used in our system. That allows 16 different measurements in the frequency range from 4 kHz to 382 kHz. However, there is a frequency bandwidth limitation due to the self-resonant frequency of the coil and a maximum current limit due to the increased circuit resistance. That is caused by the skin effect of the wires and the proximity effect that can be observed, especially in tightly wound turns of the excitation coil. 

The skin effect was reduced by using a Litz wire. In our system an excitation coil has been wound using 3.3 mm of Litz wire composed of 630 single strands with a diameter of 0.1 mm. The proximity effect between layers was decreased by a 3.5 mm air gap. This allowed for a significant increase in the coil’s self-resonant frequency and broadened the measurement frequency range.

The length of the excitation coil was selected to obtain at least 10 cm of a uniform magnetic field. It has been assumed that a region of a uniform magnetic field is the region where a deviation from the peak value is less than 10%.

A large amount of current flowing through the excitation coil resulted in a large amount of heat generated during the measurement. This additional heat production interfered with the slight temperature increase measured in calorimetric tests. An excitation coil carcass was created from a high-temperature-resistant PTFE of relatively low thermal conductivity to avoid the undesired influence on calorimetric results. The refrigerated circulator (Thermo Scientific A10, Waltham, MA USA)played the main role in the temperature stabilization of the examined sample and drained the resistive heat from the system.

### 2.2. Magnetic Probe

A schematic diagram of the excitation subsystem together with the magnetic probe is shown in Figure 3. The detection part was designed in the form of two oppositely wound receiving coils. The nanoparticle sample is placed tightly in one of the coils, which is responsible for the proper detection of the magnetization signal. The role of the second coil is to cancel the signal induced in the detection coil due to the primary magnetic field. This configuration allows for detecting the first harmonic of the magnetization signal. The voltage induced across the detection coil is then analyzed by the SR865A lock-in amplifier, which also needs a reference signal for dual-phase demodulation. The same carcass was also used to wind the reference coil, which is necessary to provide a reference signal for the lock-in amplifier. The coils were wound on the same 3D-printed carcass made from polylactic acid filament (PLA) (Figure 1b), ensuring high thermal resistance.

The length of both coils was adjusted to the length of a 2 mL sample. Keeping a distance between them was necessary in order to reduce the coils’ mutual inductance and the undesired cancellation of the magnetization signal. However, it is impossible to wind precisely two identical detection coils. Therefore, there will always be some remaining signal resulting from imperfect cancellation of the induced voltage due to the primary magnetic field. An additional positioning system was necessary to move detection coils along the longitudinal axis of the excitation coil. The linear actuator driven by the step motor enabled precise cancellation of the residual signal in the detection circuit. This procedure is essential, especially if the first harmonic of the magnetization signal is measured.

According to Faraday’s law, changing the magnetization of the sample induces a voltage across the detection coil in the following form:(1)$$V_{out}\left(t\right)=NS\mu_{0}\frac{\partial M}{\partial t}=NS\mu_{0}\chi \frac{\partial H}{\partial t}\;\;$$
where N, S, χ are the number of turns, cross-section area, and complex magnetic susceptibility, respectively. Primary AC magnetic field is expressed in the form:(2)$$H\left(t\right)=H_{0}\sin\left(\omega t\right)\;$$

Therefore lock-in amplifier is provided with the following signal:(3)$$V_{out}\left(t\right)=NS\mu_{0}\chi H_{0}\omega \cos\left(\omega t\right)\;$$

This formula is valid only if linear magnetization occurs and no other harmonics are induced in the detection circuit. The lock-in output signal is obtained by sine and cosine demodulation that isolates the signal at the target frequency and decomposes it in the form of in-phase (X) and out-of-phase (Y) components depending on $\chi^{\prime }$ and $\chi^{\prime \prime }$, respectively [12]:(4)$$X=\frac{G}{\sqrt{2}}NS\mu_{0}\chi^{\prime }H_{0}\omega \cos\left(\theta \right),$$
(5)$$Y=\frac{G}{\sqrt{2}}NS\mu_{0}\chi^{\prime \prime }H_{0}\omega \sin\left(\theta \right),$$
where G is the gain of the lock-in amplifier and θ is the phase shift between the magnetization signal and reference signal.

Measurements using lock-in phase sensitive detection require initial precise positioning of the empty carcass along the longitudinal axis of the excitation coil to achieve complete cancellation of signals induced in detection and cancellation coils that should subtract from each other. The positioning was performed with an electrically driven actuator with a precision of 0.01 mm. A nanoparticle sample was placed into the detection coil, and a magnetic field was applied. Real and imaginary parts of the complex magnetization signal were registered with and without the nanoparticle sample. This measurement was repeated several times to obtain a better signal-to-noise ratio. The lock-in amplifier was provided with the reference signal by the reference coil wound on the same carcass with receiving coils. The positioning of the detection and cancellation coils had to be repeated for each resonant frequency. The required correction was very subtle (less than one millimeter).

### 2.3. Calorimetric Probe

The performance of calorimetric tests needs the same excitation subsystem but a different sensing probe. The schematic diagram of the calorimetric setup is shown in Figure 4. 3D-printed support (Figure 1c) was designed to work with a flow-through temperature stabilization system driven by a refrigerated circulator Thermo Scientific A10. It allows for performing the measurements at a given temperature (from 4–100 °C) and drains resistive heat generated by a large amount of high alternating current. The carcass was printed with water-tight acrylonitrile butadiene styrene (ABS). However, additional post-processing using acetone fumes was needed to obtain a sufficient level of water tightness. Temperature change is tracked and recorded by fiber optic thermometer Osensa FTX-200-LUX+ with an attached probe dedicated to MRI applications. The probe was placed directly through a tiny hole made in the cap of each sample and positioned in the deepest point of the cone-shaped bottom of the test tube filled with nanoparticle solution.

For superparamagnetic nanoparticles (SP MNPs), two relaxation processes, Neel and Brown, can be considered. Thermal fluctuations rotate the magnetic moment inside the nanoparticle. The fluctuation of magnetic moments has a time constant, called the Neel’s constant, and it is determined from the formula [29]:(6)$$\tau_{N}=\tau_{0}\exp\left(KV/k_{B}T\right),$$
where K is effective anisotropy constant, V is particle volume, T is temperature, $k_{B}$ is a Boltzmann constant, and $\tau_{0}\approx 10^{-9}\;s$. The second mechanism responsible for the generation of heat is the Brown process with the relaxation constant equal to:(7)$$\tau_{B}=\frac{3V_{h}\eta }{k_{B}T},$$
where $V_{h}$ is the hydrodynamic volume of the particle [m^3^], η is the viscosity of the liquid [Pa∙s] in which the nanoparticles are immersed, $k_{B}$ is a Boltzmann constant. It consists of fluctuating the particle orientation under the influence of an external magnetic field. The effective relaxation time for a particle where both processes take place is given by the formula [30]:(8)$$\tau =\frac{\tau_{N}\tau_{B}}{\tau_{N}+\tau_{B}}.\;$$

The complex magnetic susceptibility depends on cyclic frequency *ω* and is given by the formula:(9)$$\chi =\chi^{\prime }-i\chi \prime \prime ,$$
where the real part is
(10)$$\chi^{\prime }=\frac{1}{1+{\left(\tau \omega \right)}^{2}}\chi_{0},$$
and the imaginary part is
(11)$$\chi^{\prime \prime }=\frac{\tau \omega }{1+{\left(\tau \omega \right)}^{2}}\chi_{0}.\;$$

The actual susceptibility is given by [30]:(12)$$\chi_{0}=\frac{3\chi_{i}}{\xi }\left(\coth\xi -\frac{1}{\xi }\right),$$
where:(13)$$\xi =\frac{\mu_{0}M_{d}HV}{k_{B}T},\;$$

H is magnetic field strength [A m^−1^], and $\mu_{0}$ is vacuum magnetic permeability [NA^−2^]. Initial susceptibility is given by the formula [30]:(14)$$\chi_{i}=\frac{\mu_{0}\phi M_{d}^{2}V}{3k_{B}T},$$
where ϕ is the volume fraction of magnetic cores, $M_{d}$ is the domain magnetization of a suspended particle [A m^−1^]. The power dissipation P [W m^−3^] is expressed as [30]:(15)$$P=\pi \mu_{0}\chi \prime \prime fH_{0}^{2}=\pi \mu_{0}\chi_{0}H_{0}^{2}f\frac{\omega \tau }{1+{\left(\omega \tau \right)}^{2}},$$
where $H_{0}$ is magnetic field amplitude [A m^−1^], f is frequency [Hz], ω=2πf. Time dependence of the magnetic field is assumed as follows:(16)$$H\left(t\right)=H_{0}\cos\omega t.\;$$

Equation (15) can be used to calculate the imaginary part of magnetic susceptibility, provided that the power heat is known. This can be obtained in calorimetric measurement using a temperature probe dedicated to high magnetic fields. If the difference in temperature rise is measured then, neglecting the small heat flow out of the examined sample, the power can be calculated as:(17)$$P=C_{s}\rho_{S}\frac{\Delta T}{\Delta t},$$
where $C_{s}$ is the specific heat of nanoparticle suspension [J K^−1^ kg^−1^], $\rho_{s}$ is density of suspension [kg m^−3^], ΔT is temperature rise [K], Δt is heating time [s].

For calorimetric measurements, after placing a 2 mL sample of nanoparticle colloidal solution inside the carcass, it was necessary to wait until thermal equilibrium between the environment and the sample was reached. If this condition was met, the magnetic field at the first resonant frequency was applied, and magnetic losses caused a temperature increase in the examined sample. After 3 min of measurement, the magnetic field was switched off, the temperature increase was assessed, and cooling of the sample began. Before switching to the subsequent resonant frequency, thermal equilibrium had to be reached again.

A Python script manages the measurement process, communicating with all peripheral devices, including a function generator, oscilloscope, fiber optic thermometer, and refrigerated circulator.

## 3. Results

The measurement system has been evaluated. The magnetic field’s uniformity along the excitation coil’s longitudinal axis was measured using FW Bell 5180 Gauss Meter. Magnetic flux density was presented as a function of the position relative to the center of the excitation coil in Figure 5. Measurements revealed that within the excitation coil exists a 12 cm region of the homogenous magnetic field that differs less than 10% from the maximum field value.

The detection coil’s sensitivity was tested using a small stainless-steel sample moved along the longitudinal axis of the detection coil. This coil was positioned inside the excitation coil to minimize the induced signal from the AC magnetic field. Results confirmed that detection and cancellation coils have similar sensitivity in the z-axis, allowing for the cancellation of the primary induced voltage with only a slight imbalance signal. The highest sensitivity was obtained when the sample was positioned in the region marked with red lines in Figure 5b. In this area, the maximum deviation of the magnetization signal’s sensitivity from the peak value was 22%. The detection coil’s sensitivity, defined as a ratio between induced voltage and susceptibility, ($U_{ind}/\chi$), was determined and amounted to 71.2 mV at a frequency of 10 kHz and a magnetic field value of 2.52 mT. Measurements were conducted using a small stainless-steel sample of magnetic susceptibility value (χ=1.22) positioned in the area of the maximum induced voltage.

The properties of all the coils used in the measurement system are presented in Table 1.

An example of a heating curve obtained in calorimetric measurements for 25 nm nanoparticles is presented in Figure 6a. The heating time for all the examined samples was set to 3 min. The temperature was recorded using the optic fiber probe placed in a conical cavity at the bottom of the test tube filled with colloidal solution. The imaginary part of complex magnetic susceptibility was calculated from the temperature increase (ΔT) using the formula:(18)$$\chi \prime \prime =\frac{\Delta TC_{s}\rho_{S}}{\Delta t\pi \mu_{0}fH_{0}^{2}}\;$$
derived from Equations (15) and (17).

Figure 6b presents results for AC magnetic measurements using a lock-in amplifier. The in-phase (*X*), out-of-phase (*Y*), $R=\sqrt{X^{2}+Y^{2}}$, and $\theta =tan^{-1}\left(Y/X\right)$ signals before, during, and after removing the sample from the detection coil are depicted. For this method, the imaginary part of the complex magnetic susceptibility was calculated based on out-of-phase (*Y*) signal using Equation (5).

Four samples of different diameters (15, 20, 25, and 30 nm) were tested using AC magnetometry and the calorimetric method. The test tubes with 2 mL nanoparticle colloidal solution with 5 mg/mL concentration were prepared (Figure 1d). In the experiment, Ocean NanoTech SPA nanoparticles were used, which had been previously characterized in [31,32,33]. They are prepared by the thermo-decomposition method. Each nanoparticle is a single crystal with a maghemite or magnetite structure [34].

Magnetic nanoparticles, under the influence of a radiofrequency alternating magnetic field, can generate heat as a result of susceptibility loss, hysteresis loss, and viscous heating, i.e., stirring [35]. In superparamagnetic nanoparticles (SP MNPs) the heating occurs via susceptibility loss related to two relaxation processes: Neel and Brown. In many publications [25,36,37], authors indicate that the boundary between the Neel and Brown mechanisms of relaxation is around 20 nm. The nanoparticle heating is mainly caused by Neel relaxation for NPs with a diameter below 20 nm and Brown relaxation above 20 nm. Brown and Neel relaxation constants of nanoparticles of similar core diameters were calculated and presented in [26,27]. Brown relaxation times were between $3.67\times 10^{-6}$ s for 16 nm nanoparticles and $3.13\times 10^{-5}$ s for 35 nm nanoparticles. According to Equation (11), these relaxation times correspond to peak frequencies between 43.3 and 4.3 kHz, respectively. All of these frequencies lie within our range of measurements. The Neel relaxation time was calculated only for 16 nm nanoparticles and amounted to $9.19\times 10^{-8}$ s. This value corresponds to a frequency of around 1.73 MHz, far beyond our measurements’ scope. The magnetic susceptibility peak values corresponding to Brown relaxation were observed for all the nanoparticles examined in our experiment (Figure 7).

There was a significant difference between applied magnetic fields in both magnetometry and the calorimetric measurements. In the calorimetric method, a magnetic field of 10 mT was applied to detect any reasonable temperature increase. According to the results reported in [31], a relatively low magnetic field of 2.5 mT was required to preserve the sample’s linear magnetization in AC magnetometry. This condition was checked for the magnetization signal of all nanoparticle samples using the spectrum analyzer. The alternating magnetic field with frequencies varying from 4 kHz to 382 kHz was generated using the excitation coil. Each sample was measured at 16 different resonant frequencies. The initial temperature for all experiments was set to 20 °C. Results obtained from both methods are presented on the same graphs, separately for each nanoparticle diameter (Figure 7). A result for calorimetric measurement for 15 nm at 4 kHz is missing because, within 180 s, there were no noticeable temperature changes.

Presented results for both measurement methods, in large part, proved compliance. The frequency plots of $\chi^{\prime \prime }$ are similar in shape (Figure 7). Both approaches showed that the maximum value of the imaginary part of complex magnetic susceptibility is obtained for nanoparticles with a diameter of 25 nm. Additionally, the peak value is almost at the same frequency. Nanoparticles with smaller (15 and 20 nm) and larger (30 nm) diameters heated less than 25 nm. The lowest susceptibility value was measured for nanoparticles with a diameter of 15 nm and 30 in calorimetry and magnetometry, respectively. The frequency at which the imaginary magnetic susceptibility reaches its maximum value is slightly different for the characteristics obtained by each method. These frequencies were the most consistent for 25 nm. They gained a maximum value for lower frequency in calorimetric rather than AC magnetometry measurements. The smaller the diameter, the larger the difference between the frequency value at which the maximum was reached. The best agreement between the measured values was achieved for 20 nm, whereas the characteristics of the imaginary part of complex magnetic susceptibility most deviated from each other for nanoparticles of 15 nm.

According to the linear response theory presented in [30], a maximum $\chi^{\prime \prime }$ value shifts toward lower frequencies with increasing diameter of nanoparticles. Such behavior was observed in our measurements for AC magnetometry. In our experiment, the highest magnetic losses were observed for nanoparticles of diameter 20 and 25 nm. The manufacturer claims that cores of examined nanoparticles are made from either magnetite or maghemite [34]. However, there is no information included about the ratio between both compounds. Results presented in [38] revealed that the best heating efficiency occurs for maghemite nanoparticles between 20–25 nm and magnetic between 15–20 nm. We concluded that, in our samples, the dominating magnetic compound was maghemite.

Conducted experiments showed that maximum magnetic susceptibility values are clearer to distinguish and faster to perform using AC magnetometry. Calorimetric tests are time-consuming due to the fixed heating time and time necessary to restore the thermal equilibrium before the subsequent measurement. A scenario is possible in which, at first, AC magnetometry is conducted in a wide range of frequencies. Based on the frequency at which the maximum magnetic susceptibility value is reached, it is possible to determine the narrow frequency region in which the OHF_LF_ exists. Next, the calorimetric measurements can be performed only in a limited range of frequencies. This procedure accelerates the whole experiment and allows quick verification of OHF_LF_ for each examined sample.

## 4. Conclusions

A hybrid system for the magnetic characterization of superparamagnetic nanoparticles has been developed. The system comprised a common excitation module and two replaceable probes, one for the calorimetric method and another for AC magnetometry. The proper operation of the system was confirmed. Preliminary tests were performed for nanoparticle samples of different diameters. $\chi^{\prime \prime }$ characteristics were obtained in the function of frequency using both methods. The system enables the comparison of the magnetic susceptibility curves obtained by calorimetric measurement, with the curves registered using a lock-in amplifier. Differences between the acquired characteristics will be the subject of further studies. The original procedure based on both methods was proposed. Application of this method could accelerate the estimation of OHF_LF_ compared to sole calorimetric measurements.

## Figures and Tables

**Figure 1 sensors-22-08879-f001:**
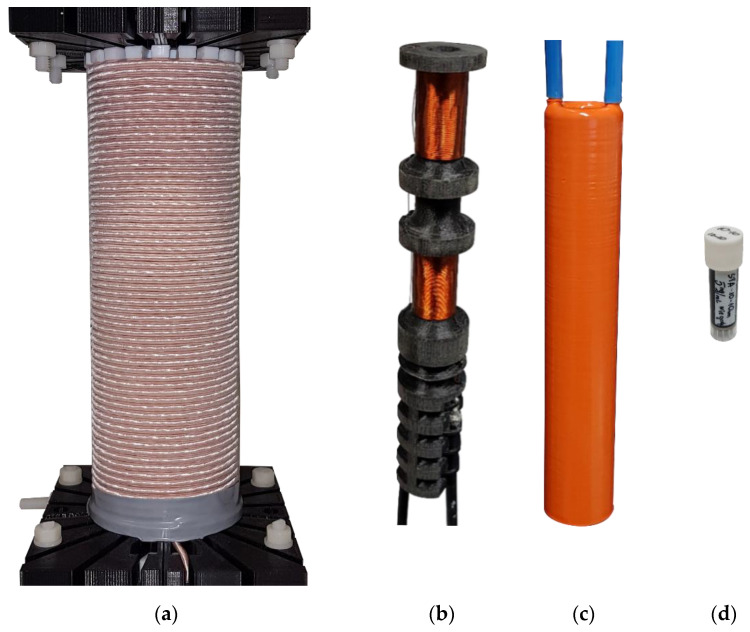
(**a**) The excitation coil (**b**) 3D-printed carcass for magnetic measurement (**c**) 3D-printed water-cooled support for calorimetric measurement (**d**) a 2 mL sample of the nanoparticle solution.

**Figure 2 sensors-22-08879-f002:**
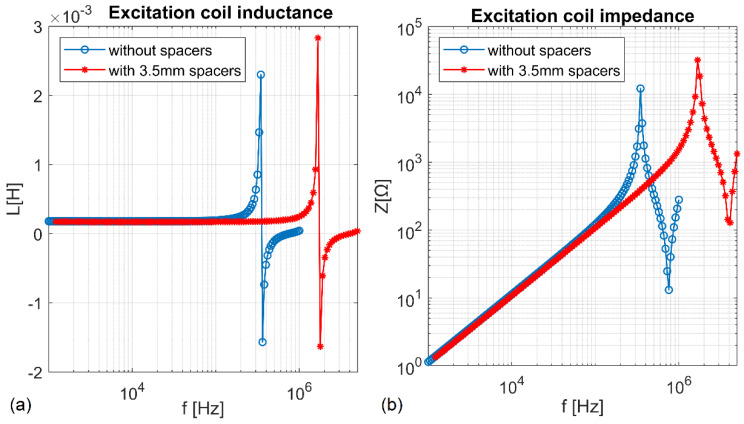
Inductance (**a**) and impedance (**b**) in a double−layer excitation coil with tightly adjacent layers (blue line) and with layers separated at a distance of 3.5 mm using wooden spacers (red line).

**Figure 3 sensors-22-08879-f003:**
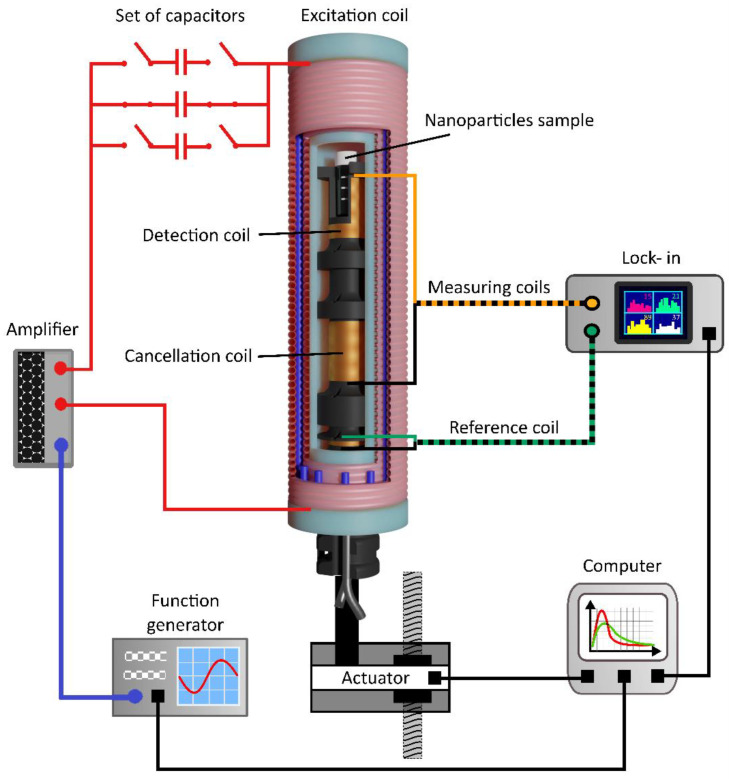
Schematic diagram of the measurement system with detection probe for AC magnetometry. The excitation module consists of the function generator, amplifier, set of different capacitors, and excitation coil. A detection probe comprises two separately wound coils and a reference coil placed on 3D printed carcass. The actuator controls the position of the probe. The lock-in amplifier detects the change in amplitude and phase of the magnetization signal.

**Figure 4 sensors-22-08879-f004:**
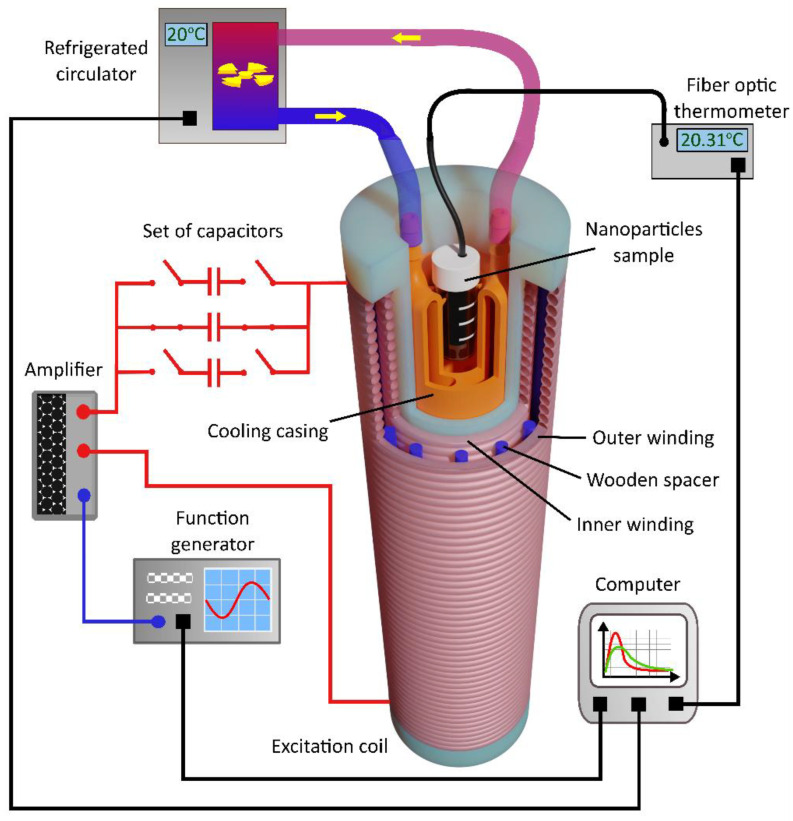
Schematic diagram of the measurement system with detection probe for calorimetric method. The excitation module consists of the function generator, amplifier, set of different capacitors, and excitation coil (the same as in the setup for magnetometry). Temperature stabilization of the sample is provided by the refrigerated circulator. The fiber optic thermometer records the temperature of the sample.

**Figure 5 sensors-22-08879-f005:**
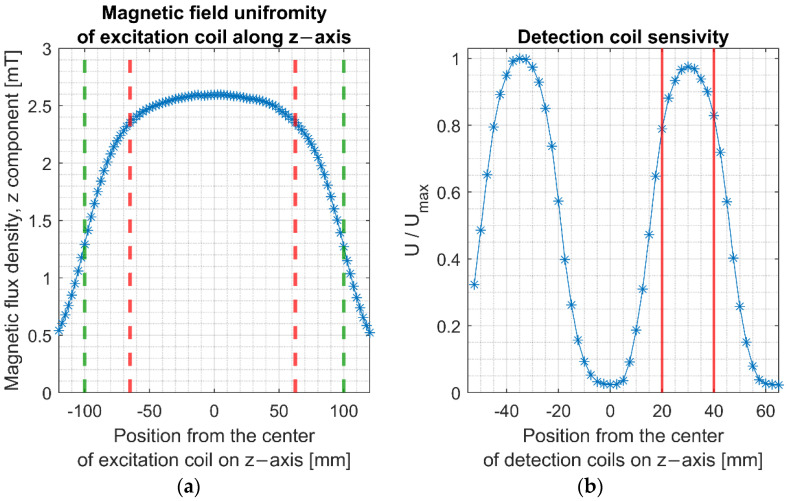
(**a**) Magnetic induction along the longitudinal axis of the excitation coil. Green dashed lines represent the physical ends of the coil, and the red dashed line shows a limited region inside the coil where the magnetic field varies less than 10% from the center point. (**b**) Relative spatial sensitivity along the longitudinal axis of two oppositely wound detection coils. The induced voltage was normalized to its maximum value. Red vertical lines show the position of a nanoparticle sample inside the detection coil.

**Figure 6 sensors-22-08879-f006:**
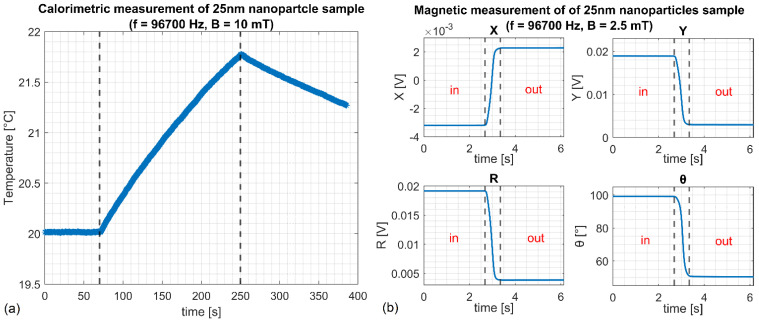
(**a**) Temperature of the nanoparticle solution during the excitation with the AC magnetic field of 10 mT. The dashed lines indicate the 3-min heating period. (**b**) Lockin output signals (X, Y, R, T), obtained for the excitation with the AC magnetic field of 2.5 mT, for the sample inside the detection coil, during and after the removal. Calorimetric (**a**) and magnetic (**b**) measurements were conducted at 96,700 Hz for a 2 mL sample of 25 nm nanoparticles.

**Figure 7 sensors-22-08879-f007:**
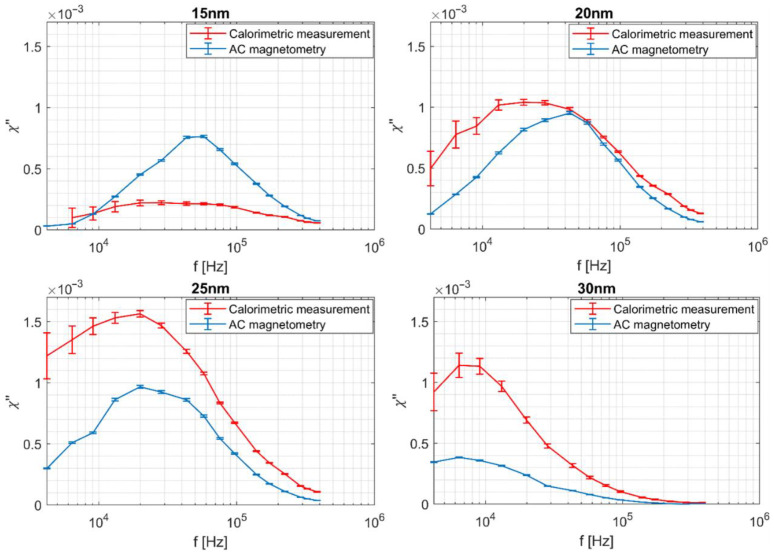
The imaginary part of complex magnetic susceptibility for different nanoparticle diameters calculated from calorimetric measurements (red line) and AC magnetometry (blue line).

**Table 1 sensors-22-08879-t001:** Properties of the coils used in the system.

	Excitation Coil	Detection Coil	Cancellation Coil	Reference Coil
Type of wire	Litz wire: 630 strands of 0.1 mm	Single strand (0.3 mm)	Single strand (0.3 mm)	Single strand (0.3 mm)
Number of layers	2	1	1	1
Number of turns per layer	61	91	91	9
Inductance [µH]	199.96	69.11	69.80	2.82
Resistance [Ω] (at 100 kHz)	0.43	1.40	1.47	0.31
Q factor (at 100 kHz)	292.0	31.0	29.8	5.7

## Data Availability

Not applicable.

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
