# Peer review of "The Hybrid System for the Magnetic Characterization of Superparamagnetic Nanoparticles"

_sensors, 2022, doi:10.3390/s22228879_

Round 1

Reviewer 1 Report

The paper presents a hybrid system of AC susceptibility magnetometry and calorimetry. The concept and its technical implementation are good. These issues need to be addressed to improve the technical quality of this paper further.

1. Page 2, Section 2 Materials and Method: The authors should explain why such a frequency range of 4 to 382 kHz of excitation frequency was selected. Particularly for the AC susceptibility measurement, the lower limit of 4kHz is too high for the Brown relaxation of particles with a diameter less than 20 nm in water viscosity. This should be compared with the particle diameter versus Neel and Brownian relaxation times, i.e., frequency. 

2. Page 5: The letter capitalization of "lock-in amplifier" and other words is inconsistent. Please perform a thorough proofread.

3. Page 5: Does the position of the detection coil needs to be changed at every measurement frequency?

4. Page 6: "Magnetic nanoparticles generate heat as a result of two relaxation processes:...". In my opinion, this sentence is inaccurate since the heat is generated dominantly due to the Neel relaxation. The increase in magnetic core temperature due to magnetic moment rotational "friction" increases the temperature around the particles.

5. Page 7, Figure 5 b): What is the detection coil's sensitivity in converting induced voltage at the detection coil to the susceptibility value?

6. Page 10: The amplitude of the excitation field for the AC susceptibility measurement was set to 2.5 mT so that the particle magnetization is in the linear region. Although the authors mentioned that the magnetization FFT spectrum confirmed this linearity response, the details, such as the manufacturer and type of the used nanoparticles, are missing. Furthermore, it would be better if the authors could include DC magnetization or reference works of the used particles.

7. Page 11, Figure 7: It will be better if the horizontal axis of Fig. 7 is plotted in the logarithmic scale.

Author Response

Dear Reviewer,

Thank you very much for the review and all the relevant remarks.

  1. Page 2, Section 2 Materials and Method: The authors should explain why such a frequency range of 4 to 382 kHz of excitation frequency was selected. Particularly for the AC susceptibility measurement, the lower limit of 4kHz is too high for the Brown relaxation of particles with a diameter less than 20 nm in water viscosity. This should be compared with the particle diameter versus Neel and Brownian relaxation times, i.e., frequency. 

We added the explanation in the Introduction and Results. We decided to choose the range of frequencies from single kHz to a few hundred kHz based on the results presented in [1][2][3]. This range covers the magnetic susceptibility peak values obtained for Brown relaxation time for magnetite/maghemite nanoparticles of diameter from 15 nm to 30 nm.

Moreover, we observed no noticeable temperature increase below 4 kHz within the heating time of our measurement (180 s). Calorimetric measurements were limited by the Osensa fiber thermometer’s sensitivity. The same lower frequency boundary was used for AC magnetometry.

  1. Lartigue, L.; Innocenti, C.; Kalaivani, T.; Awwad, A.; Sanchez Duque, M.D.M.; Guari, Y.; Larionova, J.; Gueírin, C.; Montero, J.L.G.; Barragan-Montero, V.; et al. Water-dispersible sugar-coated iron oxide nanoparticles. An evaluation of their relaxometric and magnetic hyperthermia properties. J. Am. Chem. Soc. 2011, 133, 10459–10472, doi:10.1021/ja111448t.
  2. Deatsch, A.E.; Evans, B.A. Heating efficiency in magnetic nanoparticle hyperthermia. J. Magn. Magn. Mater. 2014, 354, 163–172.
  3. Ota, S.; Takemura, Y. Characterization of Néel and Brownian Relaxations Isolated from Complex Dynamics Influenced by Dipole Interactions in Magnetic Nanoparticles. J. Phys. Chem. C 2019, 123, 28859–28866, doi:10.1021/acs.jpcc.9b06790.

  1. Page 5: The letter capitalization of "lock-in amplifier" and other words is inconsistent. Please perform a thorough proofread.

The letter capitalization of "lock-in amplifier” and the other words have been corrected.

  1. Page 5: Does the position of the detection coil needs to be changed at every measurement frequency?

Yes, it does, but the correction is very subtle (less than one millimeter). Every new measurement at another resonant frequency changes the imbalance signal and requires new positioning of the detections coils. We changed the text to reflect it better. We added information about the precision of the electrically driven actuator used in our system.

  1. Page 6: "Magnetic nanoparticles generate heat as a result of two relaxation processes:...". In my opinion, this sentence is inaccurate since the heat is generated dominantly due to the Neel relaxation. The increase in magnetic core temperature due to magnetic moment rotational "friction" increases the temperature around the particles.

To avoid misleading the readers, we have rewritten the sentence. We have added the discussion in the Results to clarify the range in which one of these heating processes dominate.

  1. Page 7, Figure 5 b): What is the detection coil's sensitivity in converting induced voltage at the detection coil to the susceptibility value?

We conducted the measurements and determined the detection coil sensitivity to the magnetic susceptibility. We added an appropriate paragraph in the text.

  1. Page 10: The amplitude of the excitation field for the AC susceptibility measurement was set to 2.5 mT so that the particle magnetization is in the linear region. Although the authors mentioned that the magnetization FFT spectrum confirmed this linearity response, the details, such as the manufacturer and type of the used nanoparticles, are missing. Furthermore, it would be better if the authors could include DC magnetization or reference works of the used particles

We added the name of the manufacturer and the product. We added references to papers that characterize nanoparticles produced by the same manufacturer. Results presented in these papers allowed to assume that excitation with the magnetic field of 2.5mT is within the linear range of magnetization.

  1. Page 11, Figure 7: It will be better if the horizontal axis of Fig. 7 is plotted in the logarithmic scale.

We changed the scale of the horizontal axis in Fig. 7 from linear to logarithmic.

Reviewer 2 Report

The authors have presented an experimental work on the design, implementation and commissioning of a AC magnetometer with the capability to measure the complex magnetic susceptibility by calorimetric and magnetic approaches. The experimental design, explanation and implementation is adequate, however, the demonstration of the function of the device is lacking a more detailed physical discussion under my point of view. Moreover, it is not clear for me the advantage claimed in the abstract and introduction about the determination of the optimal heating frequency by the combination of both measurement approaches. Due to this, I recommend a revision of the manuscript prior its publication is Sensors. Here follow my comments and concerns:

1) My principal concern is related with the hypothesis made by the authors at the beginning of the manuscript indicating that it should be better to get the complex part of the magnetic susceptibility from both methodologies to better determine the optimal heating frequency. I do not see in the discussion of the results obtained from the test samples any reference to this point which in essence is the key of the motivation of the work. A discussion about it in the framework of the results should be done.

2) Two typos in the description of the Neel and Brown relaxation times. The first one is a clarification about the viscosity units. It is written "Pas" which is Pascals times seconds, I would write it as Pa·s to emphasise it. The other is that in the Brown relaxation time (line 195), the K (anisotropy energy density) is not present as it is a purely mechanical relaxation time, and it is indicated there.

3) In the discussion of the results, when doing the comparison between calorimetric and magnetic accessed complex magnetic susceptibilities vs frequency, a physical discussion about the results related with the size of the used nanoparticles would be desirable. It is not sufficient to describe the recorded data, but it is necessary to discuss it and see if it makes sense with the nominal parameters of the test nanoparticles used.

Author Response

Dear Reviewer,

Thank you very much for the review and all the relevant remarks.

1) My principal concern is related with the hypothesis made by the authors at the beginning of the manuscript indicating that it should be better to get the complex part of the magnetic susceptibility from both methodologies to better determine the optimal heating frequency. I do not see in the discussion of the results obtained from the test samples any reference to this point which in essence is the key of the motivation of the work. A discussion about it in the framework of the results should be done. 

 We have rewritten the hypothesis. We added a description of an innovative procedure based on both magnetic and calorimetric methods. This original procedure allows for faster local optimal heating frequency determination compared to sole calorimetric measurements. It would accelerate the whole experiment and allow quick verification of the local peak in a low-frequency range (from 4 kHz to 382 kHz) for each examined sample. A paragraph discussing this approach was added to the Results.

  2) Two typos in the description of the Neel and Brown relaxation times. The first one is a clarification about the viscosity units. It is written "Pas" which is Pascals times seconds, I would write it as Pa·s to emphasise it. The other is that in the Brown relaxation time (line 195), the K (anisotropy energy density) is not present as it is a purely mechanical relaxation time, and it is indicated there. 

 Both typos have been corrected in the text.

 3) In the discussion of the results, when doing the comparison between calorimetric and magnetic accessed complex magnetic susceptibilities vs frequency, a physical discussion about the results related with the size of the used nanoparticles would be desirable. It is not sufficient to describe the recorded data, but it is necessary to discuss it and see if it makes sense with the nominal parameters of the test nanoparticles used. 

We have extended the discussion in the Results chapter by adding paragraphs concerning the physical properties of nanoparticles, including NPs size.